# Weight Loss Management and Lifestyle Changes during COVID-19 Lockdown: A Matched Italian Cohort Study

**DOI:** 10.3390/nu14142897

**Published:** 2022-07-14

**Authors:** Ramona De Amicis, Andrea Foppiani, Letizia Galasso, Angela Montaruli, Eliana Roveda, Fabio Esposito, Alberto Battezzati, Simona Bertoli, Alessandro Leone

**Affiliations:** 1International Center for the Assessment of Nutritional Status (ICANS), Department of Food Environmental and Nutritional Sciences (DeFENS), University of Milan, Via Sandro Botticelli 21, 20133 Milan, Italy; andrea.foppiani@unimi.it (A.F.); alberto.battezzati@unimi.it (A.B.); simona.bertoli@unimi.it (S.B.); alessandro.leone1@unimi.it (A.L.); 2Laboratory of Nutrition and Obesity Research, Department of Endocrine and Metabolic Diseases, IRCCS, Istituto Auxologico Italiano, 20133 Milan, Italy; 3Department of Biomedical Sciences for Health, University of Milan, Via Giuseppe Colombo 71, 20133 Milan, Italy; letizia.galasso@unimi.it (L.G.); angela.montaruli@unimi.it (A.M.); eliana.roveda@unimi.it (E.R.); fabio.esposito@unimi.it (F.E.); 4IRCCS, Istituto Ortopedico Galeazzi, Via Riccardo Galeazzi 4, 20161 Milan, Italy

**Keywords:** COVID-19 lockdown, weight loss, Mediterranean diet, chronotype

## Abstract

During the COVID-19 lockdown, lifestyle deterioration had a negative impact on weight, and yet no study has focused on patients already undergoing dietary therapy. We performed a cohort study among adults to evaluate the effect of lockdown on weight loss programs, and we investigated changes in eating habits and chronotype. We matched confined cases with non-confined cases among individuals who followed the same diet in 2017–2019. At baseline, all patients underwent a clinical examination and completed questionnaires on lifestyle. At follow-up, patients of the confined group were interviewed by a web call, and questionnaires were re-evaluated. We recruited 61 patients. The confined sample was mainly composed of middle-aged (52 (43,58) years) females (46 (75%)) with overweight (27 (44%)) or obesity (24 (39%)) and a moderate physical activity level (48 (81%)). Body weight at follow-up was significantly higher (1.1 (95% CI: 0.14, 2.1) kg) in the confined group adjusting for all matching variables. Adherence to the Mediterranean diet and eating behavior generally improved. Concerning chronotype, patients differentiated from Neither-types to Evening- and Morning-types. A well-monitored dietary therapy maintains weight loss during lockdown. Improvement in eating habits was observed; however, a shift of the circadian typology occurred.

## 1. Introduction

The first Italian case of coronavirus disease 2019 (COVID-19) was reported in the Lombardy region on 20 February 2020. The entire epidemic was characterized by a local transmission: in particular, it was very high in the regions of northern Italy, and Lombardy was the epicenter with the highest number of infections.

During the pandemic, the province of Milan (the capital city of Lombardy) witnessed a growth in the ratio between cases among its resident population and total cases in Lombardy, the former accounting for approximately 25% of the region’s total cases in the month of May 2020. Hospital bed occupancy by COVID-19 patients and the contingency measures, including total or partial lockdown, heavily restricted the healthcare resources that were diverted to home care [1].

In the home confinement experienced during the COVID-19 pandemic “Phase 1” (DPCM-GU Serie Generale n.59; 8 March 2020), lifestyle deterioration with potential negative impact on health occurred [2,3,4,5] in terms of increase in sedentary lifestyle, higher food availability, frequency of craving or eating between meals, habitual noxious foods consumption, changes in meal times and reduction in sleep quality [6]. Moreover, social distancing caused a loss of daily routine and irregular patterns of living with the risk of an increase of food addiction and binge eating behavior and an alteration of the circadian rhythm [7]. Chronotype represents interindividual differences observed in the natural propensity of behavioral manifestation relative to the light–dark cycle captured by a biological construct: the consequences of the discrepancy between social and biological time due to changes in social habits, home confinement and light–darkness cycle have been hypothesized to affect mental and physical health [7], worsening also the weight and dietary management.

These changes have been able to lead to an alarming increase in Body Mass Index (BMI) and related health risks. Any disruptions to the care of patients with obesity, including lack of access to dietary and nutritional therapies, could lead to subthreshold psychological symptoms related also to eating behavior [8] with a further detrimental worsening on diet therapy management. A recent review reported an increase in body weight (BW) in 19–49% of Italian subjects [9], and other international studies have demonstrated a higher prevalence of obesity during the COVID-19 lockdown period by coining the term “Covibesity” [9,10]. This impairment in nutritional status has made weight management more important, especially in the monitoring of weight status, eating habits and lifestyle in subjects already affected by obesity and at increased risk of infection and/or a severe course of COVID-19 [11]. The forced isolation due to pandemic containment measures abruptly also interrupted the rehabilitation programs to which many patients with severe obesity were enrolled. People affected by obesity, and especially those with severe obesity, should continue clinical rehabilitation programs, taking extra measures to avoid COVID-19 infection and reinforcing the adoption of preventive procedures, such as the use of telemedicine to maintain physician–patient communication, which was fundamental in chronic and complicated obese patients [8].

It is clear that several studies in this last year have evaluated the effect of phase 1 lockdown on eating habits, lifestyle changes and weight status and psychological aspects [2,3,4,5] thanks to telemonitoring supports, and most of them have studied the Italian population, but none have examined these aspects prospectively and in patients with obesity already undergoing a nutritional program. We therefore implemented a prospective study on a cohort of patients already following a diet therapy for obesity to evaluate the effect of lockdown on weight loss during a dietary intervention. Furthermore, as secondary outcomes, we investigated changes in eating habits, food addiction, binge eating behavior and chronotype that could contribute to a possible variation in the weight management.

## 2. Materials and Methods

### 2.1. Study Design, Setting and Participants

We performed a prospective cohort study among adults who voluntarily approached the International Center for the Assessment of Nutritional Status (ICANS), University of Milan, an urban nutritional care facility located in Milan, Italy, in order to participate in a structured weight loss program. In this study, we selected individuals who approached ICANS between October 2019 and February 2020 and who followed part of the prescribed low-calorie dietary treatment during the COVID-19 lockdown (9 March 2020–18 May 2020). Moreover, as our primary aim was to evaluate the impact of COVID-19 lockdown on weight loss, we matched confined cases with non-confined patients randomly selected among individuals who followed a low-calorie dietary treatment between January 2017 and January 2019. Confined and non-confined patients were matched for key variables affecting the main outcome (body weight at first follow-up examination): age, sex, baseline body weight, body height, baseline body fat, total daily energy intake prescribed in the diet, physical activity level (estimated as baseline weekly metabolic equivalents of task (METs)), and time to follow-up examination.

Eligibility criteria were: age ≥18 years; not pregnant and not nursing; no condition severely limiting movements and physical activity; no severe cardiovascular, neurological, endocrine, or psychiatric or eating disorders. In addition, patients in the confined group had to have followed the diet during the lockdown for at least 1 month, whereas patients in the non-confined group had to have followed the dietary treatment prior to the COVID-19 pandemic.

At baseline, patients of both groups underwent a clinical examination, anthropometric assessment and evaluation of resting energy expenditure by indirect calorimetry. In addition, they completed a battery of questionnaires designed to assess the level of physical activity, the adherence to the Mediterranean diet, the presence of binge-eating behavior and food addiction, and chronotype. Based on the nutritional evaluation, a Mediterranean hypocaloric diet was provided, and a follow-up examination was scheduled at about 3 months after the baseline evaluation with the possibility of contacting the reference dietician by call or email. At follow-up, patients of the confined group were interviewed remotely by a web call by a registered dietitian in order to obtain their current body weight, whereas non-confined patients underwent anthropometric assessment at ICANS. As we also aimed to evaluate the impact of the COVID-19 lockdown on the adherence to the Mediterranean diet, binge-eating behavior, food addiction, and chronotype, the same battery of questionnaires proposed at baseline were re-evaluated in the confined group by a web-form survey in Google Forms.

The study complied with the principles established by the Declaration of Helsinki, and written informed consent was obtained by each subject. The ethical committee of the University of Milan (n. 6/2019) approved the study procedures.

### 2.2. Variables and Measurements

Anthropometric measurements were collected at baseline for both groups and at follow-up for the non-confined group by trained registered dieticians following standard guidelines [12]. Body weight, height, waist circumference and fat mass were assessed with anthropometric methods according to standardized procedures [12,13,14]. Body weight at follow-up for the confined group was self-reported by the patients due to limitations imposed by the lockdown, but guidance was provided concerning fasted weighing, weighing in the morning, and wearing minimal clothes during weighing.

The measurement of resting energy expenditure by indirect calorimetry was performed early in the morning, after a 12 h fast, using the canopy dilution technique [15], with patients wearing a transparent ventilated canopy for 30 min, sampling gases every 30 s. Technical details of the indirect calorimeter used (Q-NRG+, Cosmed SRL, Rome, Italy) are detailed by Delsoglio et al. [16]. The Weir formula [17] was used to estimate resting energy expenditure from gas exchanges measured at rest by indirect calorimetry.

Questionnaires were self-administered at baseline to patients in both groups, whereas only individuals in the confined group completed the same questionnaires at follow-up.

Physical activity was evaluated using the Italian validated version of the International Physical Activity Questionnaire—short version (IPAQ-SF) [18]. Individuals were divided into three categories: category (1) included inactive/low-activity individuals who did not meet criteria for categories 2 or 3; category (2) considered those individuals who did 3 or more days of vigorous activity (METs ≥ 8.0) of at least 20 min per day OR 5 or more days of moderate-intensity activity (METS ≥ 4.0) or walking of at least 30 min per day OR 5 or more days of any combination of walking, moderate-intensity or vigorous intensity activities achieving a minimum of at least 600 MET-min/week were considered moderate; category (3) considered those who do vigorous-intensity activity on at least 3 days and accumulate at least 1500 MET-minutes/week OR 7 or more days of any combination of walking, moderate-intensity or vigorous intensity activities, achieving a minimum of at least 3000 MET-minutes/week, which was considered high.

Adherence to the Mediterranean diet was assessed using Mediterranean Diet Adherence Screener (MEDAS), which is a validated 14-item questionnaire used to obtain the Mediterranean score (MED score) [19]. Participants with a MED score ≥9 points were considered as complying with a dietary pattern in accordance with the Mediterranean dietary pattern.

The presence of food addiction and binge eating behavior were assessed using, respectively:-The Yale Food Addiction Scale (YFAS) [20], a validated measure of addictive-like eating behavior based upon the diagnostic criteria for substance dependence, that investigates the addiction in relation to some high-fat and high-carbohydrate foods that leads to clinically significant impairment or distress on several areas of functioning; it has two scoring options: (a) a continuous score—symptom count—indicating the number of symptoms of food addiction symptoms that have been met and (b) a diagnostic score that provides a diagnosis of food dependence when the subject presents at least three symptoms and reports clinically significant impairment and/or distress.-The Binge Eating Scale (BES) [21], 16 forced-choice questions, each with a set of 3 or 4 answers. The BES gives a score ranging from 0 to 46. Participants with a BES score >18 were identified as binge eaters [22,23].

A shortened 5-item version (rMEQ) of the standard 19-item Morningness–Eveningness Questionnaire (MEQ) was used to assess chronotype [24]. These five items assess sleeping and waking hours, peak time, morning alertness, and self-assessment of chronotype. Each item corresponds to a score: the sum of the scores allows obtaining the final rMEQ score, which varies from < 12 (Evening-types, more active in the last part of the day = eveningness) to > 17 (Morning-types, who feel wide awake and fresh in the morning = morningness). Intermediate scores were associated with Neither-types (12–17 points), who have intermediate preference.

### 2.3. Nutritional Intervention

After baseline nutritional assessment, patients were prescribed a low-calorie Mediterranean-style diet. The diet provided calories in an amount equal to the calories required for resting energy expenditure. Calories were 45–55% from carbohydrates, 25–30% from lipids, and 10–15% from protein. The diet also provided 25–30 g/day of fiber and met recommended micronutrient requirements thanks to the Mediterranean pattern (olive oil as the main cooking fat, fruits and vegetables > 4 servings/day, legumes ≥ 3 servings/week, fish/seafood ≥ 3 servings/week, use of wholemeal cereals).

### 2.4. Statistical Methods

Most continuous variables did not follow a normal distribution, and all are reported as within the median and interquartile range. Categorical variables are reported as counts and proportions. The matching of confined patients with historical controls was performed using coarsened exact matching, and the covariate balance between groups was evaluated calculating the standardized mean differences across imputations. Differences in the main outcome between confined and non-confined groups were tested using linear regression models adjusted for baseline measurement of the outcome and all matching variables (age, sex, baseline body weight, body height, baseline body fat mass, total daily energy intake prescribed, baseline weekly METs, time to follow-up examination). Matching was considered using CEM-related weights.

Secondary endpoints were tested in subgroups of the main sample, and comparisons of baseline characteristics of included and excluded patients were carried out. We sought to highlight the association between continuous scores of the questionnaires used to evaluate the secondary endpoints and time in lockdown, controlling for baseline scores, age, sex, and baseline body mass index. Linearity was not assumed for all continuous variables that were transformed using restricted cubic splines with knots at the median and boundary knots at the 10th and 90th percentiles.

Comparisons between groups were carried out using the Wilcoxon rank sum test for continuous variables and the Pearson’s Chi-squared or the Fisher’s exact test for categorical variables. We used multivariate imputation by chained equations to impute missing data in both confined and non-confined patients. Loss to follow-up was addressed as a possible source of bias comparing baseline characteristics between follow-up status groups of confined and non-confined patients.

## 3. Results

### 3.1. Participants and Descriptive Data

A diagram of patients at each stage of the study is shown in Figure 1.

The matching procedure pruned a total of 23/84 (27%) patients from the overall sample, while a total of 61/84 (73%) patients were matched. The COVID-19 pandemic caused a higher dropout to the second appointment in the confined group (48% versus 37% in the non-confined group, *p* < 0.001): patients who refused follow-up in the confined group were older (52 (43,58) versus 49 (38,58) years, *p* = 0.005) and probably not comfortable with the use of telemedicine.

Table 1 shows baseline sample characteristics for the confined and non-confined group and covariate balance on matching variables. Our exposed sample was mainly composed of middle-aged (52 (43,58) years) females (46 (75%)), doing sedentary work (33 (54%)), and married (35 (58%)). Most patients were either overweight (27 (44%)) or obese (24 (39%)), with increased waist circumference (median 98 cm (86,106)), and moderate physical activity level (48 (81%)). The most prevalent conditions were essential hypertension (15 (25%)) and dyslipidemias (14 (23%)). For the missing data, see Appendix A. Patients were in lockdown for at least 1 month with a median lockdown duration of 1.64 months (1.28, 1.94).

Table 2 shows the summary characteristics of diets of the confined and non-confined groups. The median daily calorie deficit was 580 kcal (500, 709), with adequate protein intake according to the recommended dietary allowance (@larn), body composition, and weight target (0.87 g/kg [0.80, 0.98]), averaging 50% (49, 52) calories from carbohydrates, 29% (28, 30) of calories from fats, and providing 16.7 g/1000 kcal (15.4, 17.8). The diets were generally comparable, with some statistically significant differences that we deem not clinically relevant. Most importantly, as a result of matching pre-processing, the caloric deficit was not different in the two groups.

### 3.2. Outcome Data, Main Results, and Other Analyses

#### 3.2.1. Primary Outcome

Table 3 shows unadjusted and baseline-adjusted differences in the outcome (body weight at follow-up) between exposure groups. After adjusting for baseline body weight, body weight at follow-up was significantly higher (on average 1.1 [95% CI: 0.14, 2.1] kg) in the confined group. The results were confirmed both adjusting for all matching variables and in a sensitivity analysis of imputed data.

#### 3.2.2. Secondary Outcomes

Secondary endpoints were not available for all patients included in the main analysis, so we compared included and excluded patients for each secondary endpoint. Patients included in secondary endpoint analyses were 37 and generally younger although without reaching statistical significance (51 years (42, 54) vs. 54 years (50, 58), *p* = 0.088).

Table 4 shows paired differences of questionnaire scores before and after lockdown. Table 5 shows results from the predictive models of questionnaire scores at follow-up. Adherence to the Mediterranean diet improved (+ 1.8 points [1.1, 2.4] of MEDAS score) while remaining in the moderate adherence group. YFAS (−1.1 [−2.2, −0.02]) and BES scores (−3.2 [−4.7, −1.6]) generally improved with reductions in both food addiction and binge eating behavior, although baseline levels were not indicative of overt food addiction or binge eating. rMEQ categories were significantly different, with patients’ differentiating evenly from Neither-types to Evening- and Morning-types (12 Neither-types patients converted to six Evening- and six Morning-type). BES and rMEQ scores at follow-up were significantly associated with time in lockdown, and their modeled relationship is shown in Figure 2: more time spent in lockdown was associated with slightly higher BES scores and slightly lower rMEQ scores (*p* = 0.027 and *p* = 0.038, respectively).

## 4. Discussion

To our knowledge, this is the first longitudinal study analyzing the effect of home confinement during the COVID-19 lockdown on the weight loss of a well-matched cohort of patients undergoing a dietary intervention for obesity.

After adjusting for age, sex, body weight and height at baseline, body fat mass at baseline, total daily prescribed energy intake, physical activity at baseline, and time to follow-up examination, patients in lockdown monitored by telemedicine maintained the weight loss trend as the control group followed in a clinical setting; however, the confined group reported a body weight at follow-up significantly higher (on average 1.1 (95% CI: 0.14, 2.1) kg). Other studies on the Italian population reported a general increase in BMI and a worsening of nutritional status during lockdown [2,3,5,9], while in our sample, dietary intervention was still effective in weight management, although at a reduced rate in comparison to pre-pandemic interventions, which was probably due to the dietary therapy course undertaken before home confinement that may have led to a previous improvement in eating habits. Analysis of the secondary outcomes showed that adherence to the Mediterranean diet improved during lockdown: in fact, adherents to the Mediterranean diet during the lockdown increased from 14 to 46%. It is likely that the nutritional advice prescribed during the first visit, followed by more time to organize and prepare food thanks to the reduction in the work-related activities, was protective against the weight gain. A similar pattern was also shown in Italian surveys on eating habits during the COVID-19 lockdown [3,4], in which a group of respondents of northern Italy reported greater consumption of fresh foods, especially fruit and vegetables, but also an increased habit of breakfast consumption. So, the reduced rate of weight loss in confined patients was reasonably due to reduced physical activity, which was a constraint imposed by the lockdown condition. Considering the general deterioration of nutritional status recorded in the general population [2,3,4,5], weight loss intervention could be seen as a protective factor during isolation to prevent those changes. A greater focus on physical activities that are feasible during isolation could prevent even more nutritional status deterioration or improve weight loss intervention.

Moreover, the Mediterranean pattern of the dietary prescription could explain the improvement of food addiction and binge eating behavior observed in our confined group: in fact, a previous study already showed that adherence to the Mediterranean diet was inversely related to binge eating disorder in patients seeking a weight loss program [25], in particular in the middle-aged group, even if the specific mechanisms by which a better adherence to the Mediterranean diet could influence the eating behavior are unknown. Additionally, more time in meal preparation and sharing, with also more organization and care of the meal space, may have allowed for an increase in mindful eating, even if unconsciously. Mindful eating, in fact, is suggested as an essential strategy for controlling food addiction and binge eating behavior [26,27] and consists of paying attention and devoting proper time to chewing food thoroughly before swallowing, drinking water between bites of food, savoring the aroma and taste of food, turning off the television and computer while eating, and creating a pleasurable experience around food and eating [28]. The greater time availability of lockdown allowed all that in contrast to the times punctuated by the work routine.

In the secondary outcomes analysis of the weight loss management, patients in lockdown showed a shift in their circadian typology, differentiating evenly from Neither-types to Evening- and Morning-types, with a modification of their own sleep–wake cycle in the absence of work-related activities and social commitments. This change resulted in a higher prevalence of Evening- and Morning-types at follow-up: Evening-type individuals are mostly engaged with unhealthy eating habits related to obesity and therefore have been hindered in the case of weight loss interventions [29], and this could partially explain the lower weight loss in our confined group. On the other hand, Morning-type individuals are mostly engaged with healthy and regular eating habits, and this could partially explain the higher adherence to the Mediterranean diet during lockdown.

Nonetheless, binge eating behavior and chronotype at follow-up were significantly associated with time in lockdown: more time spent in lockdown was associated with slightly higher BES scores and slightly lower rMEQ scores. Social distancing and home confinement has been associated with disordered eating in cross-sectional studies [30]: emotional distress, boredom and social isolation lead to overeating and disordered eating. Moreover, the mandatory COVID-associated isolation affected sleep timing and duration [31]: subjects slept longer and later in the absence of work-related activities, which was better aligned with their internal time; however, a significant delay in sleep timing results in increasing the later chronotypes, while a significant advance in sleep timing results in increasing the early chronotype [32,33].

All these results could confirm how the determinants of weight loss are multiple: while a low-calorie diet could maintain the weight loss trend in the medium term even in an abnormal situation such as home confinement during the COVID-19 lockdown, overall lifestyle plays a crucial role in the proportion of this trend. Lifestyle includes energy balance, psychological and behavioral determinants, dietary pattern and time to spend on the weight loss program and on monitoring weight and eating [34], and each of them should be considered in the management of the emerging “Covibesity” [10], eventually through telemedicine. In fact, even the use of telemedicine may have prevented weight regain and maintenance of the weight loss trend in our study. Telemedicine interventions could provide a safe, remote alternative and may expand treatment access to hard-to-reach patients. In particular, video conferencing can be particularly useful in capturing the bond associated with in-person care [35], strengthening the trust between healthcare providers and the patient, especially in the field of nutrition, where the nutritionist needs to be able to comprehend the patient’s daily routine to help with grocery shopping, weekly menu planning, meal management, and food-related emotionality with personalized recommendations required by the modern nutritional therapies [36].

The first strength of this study is that it was a well-matched cohort study with a prospective design with enough controls relative to cases (6:1 allocation) to have good outcomes accuracy. Moreover, the prescribed diets were based on resting energy expenditure measured by a gold standard method as indirect calorimetry. However, we must consider some limitations: recruitment was completed before lockdown and could have caused an increase in dropouts at follow-up (about 50%), which was probably due to the use of telemedicine as monitoring that led older or less responsive subjects to drop out of the weight loss program. The follow-up through telemonitoring was conducted at 3 months, which is a relatively long time considering the immediate effects of a low-calorie diet on weight; however, during these 3 months, there were contacts via chats or e-mail with the reference dietician. In addition, the lower weight loss of the confined group could also depend on the possible uncertainty of self-reported results: only reported body weight was evaluated at follow-up in confined patients because of home confinement due to the pandemic. Lastly, lifestyle was studied in a subgroup: the use of online questionnaires led to lower compliance in completing the survey and a consequently lower number of respondents.

## 5. Conclusions

Despite the expected and reported lifestyle deterioration during lockdown, a well-monitored weight loss program had a protective effect against weight gain. Improvement in the Mediterranean dietary pattern and in food addiction and binge eating behavior was observed, which was probably also due to the use of telemedicine, which may have strengthened the link between healthcare providers and patients during such a difficult time. Lastly, a shift toward extreme chronotypes occurred, which was probably due to the deep interruption of work and social activities.

## Figures and Tables

**Figure 1 nutrients-14-02897-f001:**
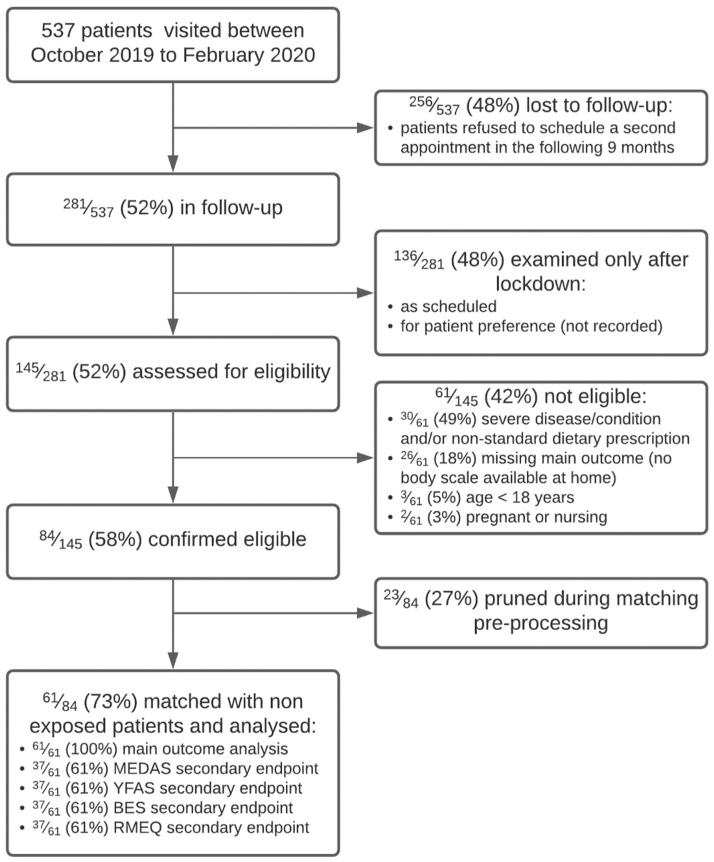
***CONSORT*** participant flow diagram.

**Figure 2 nutrients-14-02897-f002:**
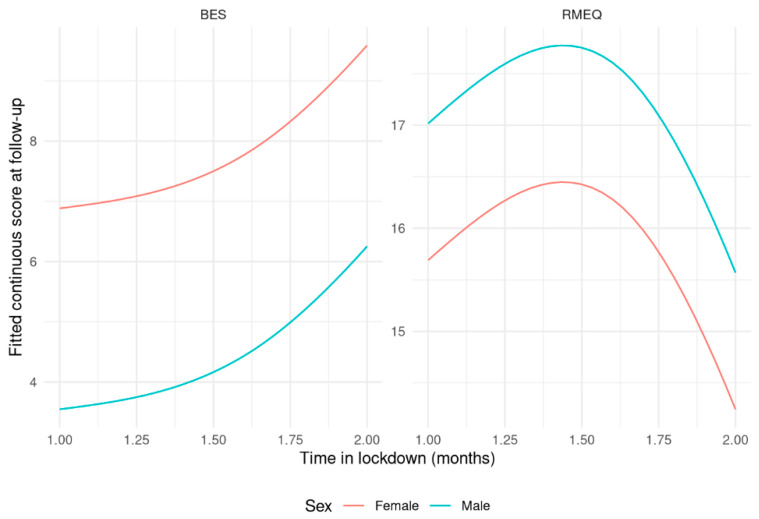
Partial effects of questionnaire scores at follow-up against time in lockdown. The left panel shows Binge Eating Scale score and the right panel shows the short Morningness–Eveningness Questionnaire score.

**Table 1 nutrients-14-02897-t001:** Baseline characteristics of matched confined and non-confined group, with covariate balance.

Characteristics	N	Confined, N = 61 ^1,2^	Not Confined, N = 402 ^1,2^	SMD across Imputations ^3^
Age (years)	463	52 (43, 58)	53 (48, 58)	−0.03 (−0.03, −0.02)
Sex	463			0 (0, 0)
Female		46 (75%)	340 (85%)	
Male		15 (25%)	62 (15%)	
Education	463			
Primary		0 (0%)	6 (1.5%)	
Lower secondary		1 (1.6%)	29 (7.2%)	
Upper secondary		29 (48%)	190 (47%)	
Tertiary		1 (1.6%)	12 (3.0%)	
Bachelor		29 (48%)	157 (39%)	
Other		1 (1.6%)	8 (2.0%)	
Occupation	463			
Unemployed		1 (1.6%)	15 (3.7%)	
Student		0 (0%)	5 (1.2%)	
Homemaker		2 (3.3%)	12 (3.0%)	
Retired		6 (9.8%)	40 (10.0%)	
Laborer		3 (4.9%)	12 (3.0%)	
Office		33 (54%)	187 (47%)	
Freelancer		2 (3.3%)	47 (12%)	
Other		14 (23%)	84 (21%)	
Marital status	460			
Single		21 (35%)	118 (30%)	
Married		35 (58%)	229 (57%)	
Widowed		0 (0%)	13 (3.2%)	
Divorced		4 (6.7%)	40 (10%)	
Body height (m)	463	1.64 (1.59, 1.69)	1.62 (1.58, 1.68)	−0.06 (−0.07, −0.05)
Body weight (kg)	463	75 (68, 91)	73 (66, 83)	−0.03 (−0.03, −0.02)
Body mass index (kg/m²)	463	28.7 (25.9, 32.8)	27.9 (25.7, 30.8)	
Body mass index category	463			
Underweight		1 (1.6%)	3 (0.7%)	
Normal weight		9 (15%)	76 (19%)	
Overweight		27 (44%)	191 (48%)	
Obese		24 (39%)	132 (33%)	
Waist circumference (cm)	459	98 (86, 106)	95 (88, 103)	
Unknown		0	4	
Body fat (as %)	440	39.8 (34.7, 43.3)	41.5 (37.8, 43.9)	−0.06 (−0.08, −0.04)
Unknown		2	21	
Resting energy expenditure (kcal/day)	461	1416 (1268, 1666)	1332 (1233, 1459)	
Unknown		1	1	
Metabolic equivalents of task (MET-minutes/week)	448	1059 (884, 1377)	1215 (885, 2120)	−0.06 (−0.12, −0.01)
Unknown		2	13	
Physical activity level	448			
Low		5 (8.5%)	66 (17%)	
Moderate		48 (81%)	242 (62%)	
High		6 (10%)	81 (21%)	
Unknown		2	13	
Prescribed energy intake (kcal/day)	460	1450 (1300, 1700)	1350 (1250, 1500)	0.07 (0.07, 0.08)
Unknown		0	3	
Diet duration (months)	463	2.56 (2.50, 4.34)	2.76 (2.53, 3.22)	−0.07 (−0.07, −0.06)
Time in lockdown (months)	463	1.64 (1.28, 1.94)	0.00 (0.00, 0.00)	

^1^ Median (IQR); n (%). ^2^ Unweighted matches across imputations. ^3^ SMD = standardized mean difference; Mean (Range).

**Table 2 nutrients-14-02897-t002:** Dietary composition of the prescribed diets in the confined and non-confined groups.

Characteristics	Confined, N = 61 ^1^	Not Confined, N = 402 ^1^	*p*-Value ^2^
Caloric deficit (kcal/day)	580 (500, 709)	563 (458, 725)	0.5
Protein (g)	68 (62, 77)	65 (60, 71)	0.002
Protein (g/kg body weight)	0.87 (0.80, 0.98)	0.86 (0.81, 0.92)	0.3
Carbohydrate (g)	197 (178, 233)	188 (174, 207)	0.055
Carbohydrate fraction of energy intake (as %)	49.60 (48.52, 51.54)	50.31 (49.26, 51.50)	0.069
Fat (g)	50 (42, 57)	47 (43, 54)	0.6
Fat fraction of energy intake (as %)	28.87 (27.71, 30.39)	30.38 (29.05, 31.69)	<0.001
Fibers (g)	25.4 (22.3, 28.0)	25.2 (23.0, 27.2)	0.6
Fibers (g/1000 kcal energy intake)	16.71 (15.40, 17.78)	17.57 (16.28, 18.93)	<0.001

^1^ Median (IQR). ^2^ Wilcoxon rank sum test.

**Table 3 nutrients-14-02897-t003:** Differences of body weight at follow-up between confined and non-confined groups both unadjusted and adjusted for baseline measurement of the outcome and for baseline measurement of the outcome and all matching variables (age, sex, baseline body weight, body height, baseline body fat mass, total daily energy intake prescribed, baseline weekly METs, time to follow-up examination).

Characteristics	Confined ^3^	Not Confined ^3^	Unadjusted ^1^			Adjusted ^2^		
			Difference	95% CI	*p*-Value	Difference ^4^	95% CI ^4^	*p*-Value ^4^
Body weight at follow-up (kg)	77.0 (17.1)	76.4 (17.0)	0.58	−4.1, 5.2	0.8	1.1	0.14, 2.1	0.025

^1^ ANOVA. ^2^ ANCOVA, adjusted for body weight at baseline. ^3^ Mean (SD). ^4^ CI = Confidence Interval.

**Table 4 nutrients-14-02897-t004:** Differences of questionnaires scores before and during lockdown.

Characteristics	N	Before Lockdown ^1^	During Lockdown ^1^	Difference	95% CI ^2^	*p*-Value ^3^
Mediterranean Adherence Screener						
Continuous score	37	7.00 (5.75, 8.00)	8.00 (7.00, 9.00)	1.8	1.1, 2.4	<0.001
Categorical score	37					0.001
Not adherent		32 (86%)	20 (54%)			
Adherent		5 (14%)	17 (46%)			
Yale Food Addiction Scale						
Continuous score	37	1.00 (0.00, 2.25)	0.00 (0.00, 1.00)	−1.1	−2.2, −0.02	0.047
Categorical score	37					0.4
No food addiction		34 (92%)	36 (97%)			
Mild food addiction		1 (2.7%)	0 (0%)			
Moderate food addiction		1 (2.7%)	1 (3.0%)			
Severe food addiction		1 (2.7%)	0 (0%)			
Binge Eating Scale						
Continuous score	37	8.0 (7.0, 12.0)	6.0 (2.0, 10.0)	−3.2	−4.7, −1.6	<0.001
Categorical score	37					0.2
Minimal binge eating problems		35 (95%)	37 (100%)			
Moderate binge eating problems		1 (2.5%)	0 (0%)			
Severe binge eating problems		1 (2.5%)	0 (0%)			
Reduced Morningness–Eveningness Questionnaire						
Continuous score	37	14.00 (13.00, 15.00)	15.00 (13.00, 17.00)	0.36	−0.89, 1.6	0.6
Categorical score	37					0.001
Evening-types		1 (2.5%)	7 (19%)			
Neither-types		35 (95%)	23 (62%)			
Morning-types		1 (2.5%)	7 (19%)			

^1^ Median (IQR); n (%). ^2^ CI = Confidence Interval. ^3^ Paired *t*-test; random intercept logistic regression.

**Table 5 nutrients-14-02897-t005:** Coefficients of linear models for questionnaire continuous scores at follow-up. For each model, the outcome was the continuous scores of the questionnaire used to evaluate the secondary endpoints and predictors time in lockdown, baseline score, age, sex, and baseline body mass index.

Characteristics	MEDAS ^1^			YFAS ^2^			BES ^3^			rMEQ ^4^		
	Beta	95% CI ^5^	*p*-Value	Beta	95% CI ^5^	*p*-Value	Beta	95% CI ^5^	*p*-Value	Beta	95% CI ^5^	*p*-Value
(Intercept)	6.5	4.8, 8.3	<0.001	0.30	−1.8, 2.4	0.8	3.1	−0.93, 7.1	0.13	16	12, 19	<0.001
Age spline ^6^												
Below median	−0.54	−2.7, 1.6	0.6	−0.09	−2.3, 2.1	>0.9	0.95	−3.9, 5.8	0.7	3.7	−0.55, 7.9	0.085
Above median	0.25	−1.1, 1.6	0.7	−0.12	−1.9, 1.6	0.9	2.7	−0.50, 5.9	0.10	−2.7	−5.6, 0.25	0.071
Sex												
Female												
Male	1.1	−0.16, 2.4	0.083	−0.11	−1.9, 1.7	0.9	−3.3	−6.6, −0.08	0.045	1.3	−1.8, 4.4	0.4
Body mass index spline ^6^												
Below median	1.4	−1.4, 4.1	0.3	−1.5	−5.4, 2.4	0.4	−2.2	−8.5, 4.1	0.5	−4.0	−9.5, 1.5	0.15
Above median	0.81	−0.21, 1.8	0.11	−0.72	−2.4, 1.0	0.4	0.83	−1.6, 3.3	0.5	−2.3	−4.6, −0.13	0.039
Baseline score spline ^6^												
Below median	1.3	−1.4, 4.0	0.3	3.5	0.64, 6.3	0.021	5.3	0.91, 9.7	0.020	−1.8	−6.7, 3.1	0.5
Above median	0.71	−0.81, 2.2	0.3	1.6	0.44, 2.7	0.012	4.7	2.6, 6.8	<0.001	0.74	−1.9, 3.3	0.6
Time in lockdown spline ^6^												
Below median	1.4	−1.4, 4.2	0.3	−0.36	−3.6, 2.9	0.8	3.7	−3.2, 11	0.3	−1.1	−7.1, 4.8	0.7
Above median	0.83	−1.2, 2.9	0.4	0.89	−1.2, 3.0	0.4	4.8	0.60, 9.0	0.027	−5.3	−10, −0.33	0.038

^1^ MEDAS = Mediterranean Adherence Screener. ^2^ BES = Binge Eating Scale. ^3^ YFAS = Yale Food Addiction Scale. ^4^ rMEQ = Reduced Morningness–Eveningness Questionnaire. ^5^ CI = Confidence Interval. ^6^ Restricted cubic spline with knots at the median and boundary knots at the 10th and 90th percentiles.

## Data Availability

All data generated or analyzed during this study are included in this article. Further enquiries can be directed to the corresponding author.

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
