# Peer review of "Weight Loss Management and Lifestyle Changes during COVID-19 Lockdown: A Matched Italian Cohort Study"

_nutrients, 2022, doi:10.3390/nu14142897_

Round 1

Reviewer 1 Report

Dear authors this is a very interesting study, which aim at identifying the effect of COVID confinement to body weight and lifestyle parameters among persons attending a weight loss management program in Italy. 

The introduction is well-written providing the necessary information. Throughout the text I suggest the change of the wording exposed/not exposed to confined/not confined. I was confused in the beginning of the review procedure.

Methodology is adequately described. I would suggest in lines 178-188 to change the presentation of the physical activity categories to 1, 2, 3. Furthermore, please change discrete variables to categorical.

In line 277 essential hypertension is mentioned. What do you mean by that?

I suggest that when presenting data at the Tables you should include p values for all variables, either the differences are significant or not.

Finally the text would be improved by an english editing.

Author Response

  • Dear authors this is a very interesting study, which aim at identifying the effect of COVID confinement to body weight and lifestyle parameters among persons attending a weight loss management program in Italy. The introduction is well-written providing the necessary information.
    • We really thank the reviewer for this valuable consideration.
  • Throughout the text I suggest the change of the wording exposed/not exposed to confined/not confined.
    • Agree, we have replaced all occurrences of “exposed”/”not exposed” to “confined”/”not confined”.
  • I would suggest in lines 178-188 to change the presentation of the physical activity categories to 1, 2, 3.
    • Agree, done.
  • Furthermore, please change discrete variables to categorical.
    • Agree, done.
  • In line 277 essential hypertension is mentioned. What do you mean by that?
    • Primary or idiopathic hypertension in which secondary causes are not present and that is typically associated with obesity, insulin resistance, high salt intake, and other factors (https://doi.org/10.1161/01.CIR.101.3.329).
  • I suggest that when presenting data at the Tables you should include p values for all variables, either the differences are significant or not.
    • We report p-values in all tables besides Table 1, where covariate balance between groups is reported as standardized mean difference. P-values from hypothesis testing were intentionally not reported in Table 1 as this reflect best methodological practice when reporting covariate balance after matching (https://doi.org/10.1093/pan/mpl013 https://doi.org/10.1093/pan/mpl013 https://doi.org/10.1080/00273171.2011.568786 https://doi.org/10.1002/sim.3697 https://doi.org/10.1080/00273171.2011.540475 https://doi.org/10.1016/j.jclinepi.2014.08.011). Quoting: “p-values from hypothesis testing are influenced by sample size, which fluctuates during adjustment, and the theory behind them is inappropriate because balance is a quality solely of the sample in question, not in relation to a population. The relevant information in a hypothesis test for group differences is the standardized magnitude of the group difference, and so such a measure is preferred.” (https://CRAN.R-project.org/package=cobalt ).
  • Finally the text would be improved by an english editing.
    • Thank you for the suggestion. We have made several changes throughout the manuscript to improve english spelling and grammar.

Reviewer 2 Report

In this work, De Amicis et al. present a series of very interesting data related to following a Mediterranean diet and its effects on body weight and other parameters during the Covid-19 lockdown. In the opinion of this reviewer, it is necessary to address some points. Such points are: 1) better describe the results obtained, the description is very general; 2) in the methodology it is necessary to describe the procedures without redounding; 3) the figures and tables require a more detailed description, since the legends should help their self-description; 4) considering the quantity of results, a richer discussion is necessary; 5) finally, it would be interesting to include, as supplementary material, the questionnaires used and cite them in the methodology section.

Author Response

  • In this work, De Amicis et al. present a series of very interesting data related to following a Mediterranean diet and its effects on body weight and other parameters during the Covid-19 lockdown. In the opinion of this reviewer, it is necessary to address some points.
    • We thank the reviewer for this valuable consideration and the thorough review.
  • better describe the results obtained, the description is very general;
    • We have expanded both the descriptive part (see pages 7,8) and the part on primary and secondary outcomes (see page 8)
  • in the methodology it is necessary to describe the procedures without redounding;
    • Agree, we have simplified several paragraph in the “Materials and Methods” section (see pages 4,5).
  • the figures and tables require a more detailed description, since the legends should help their self-description;
    • We have made all tables and figures self-descriptive, expanded titles and tables
  • considering the quantity of results, a richer discussion is necessary;
    • We thank the reviewer for this suggestion. We have expanded both the concise summary of the investigated problem (see page 9) and the critical discussion of major and minor findings (see page 10).
  • finally, it would be interesting to include, as supplementary material, the questionnaires used and cite them in the methodology section.
    • Fixed it. See the supplementary materials in which we added all the questionnaires used.

Round 2

Reviewer 1 Report

Dear authors thank you for the revised manuscript.